# Persistent Organic Pollutants in Food: Contamination Sources, Health Effects and Detection Methods

**DOI:** 10.3390/ijerph16224361

**Published:** 2019-11-08

**Authors:** Wenjing Guo, Bohu Pan, Sugunadevi Sakkiah, Gokhan Yavas, Weigong Ge, Wen Zou, Weida Tong, Huixiao Hong

**Affiliations:** U.S. Food & Drug Administration, National Center for Toxicological Research, 3900 NCTR Road, Jefferson, AR 72079, USA; Wenjing.Guo@fda.hhs.gov (W.G.); Bohu.Pan@fda.hhs.gov (B.P.); Suguna.Sakkiah@fda.hhs.gov (S.S.); Gokhan.Yavas@fda.hhs.gov (G.Y.); Weigong.Ge@fda.hhs.gov (W.G.); Wen.Zou@fda.hhs.gov (W.Z.); Weida.Tong@fda.hhs.gov (W.T.)

**Keywords:** persistent organic pollutants, food contamination, human health, food safety, environmental contaminants

## Abstract

Persistent organic pollutants (POPs) present in foods have been a major concern for food safety due to their persistence and toxic effects. To ensure food safety and protect human health from POPs, it is critical to achieve a better understanding of POP pathways into food and develop strategies to reduce human exposure. POPs could present in food in the raw stages, transferred from the environment or artificially introduced during food preparation steps. Exposure to these pollutants may cause various health problems such as endocrine disruption, cardiovascular diseases, cancers, diabetes, birth defects, and dysfunctional immune and reproductive systems. This review describes potential sources of POP food contamination, analytical approaches to measure POP levels in food and efforts to control food contamination with POPs.

## 1. Introduction

Chemical contaminants have become a concern in terms of food safety due to pesticide residue and environmental contaminants detected in the food supply. A large amount of pollutants generated from rapidly developing agricultural and industrial sectors have been released to the environment and found their way into the food supply. Given the widespread occurrence of chemical contaminants in foodstuff and their serious health risks, the prevention of food contamination is a public health priority. In recent decades, there has been a focus on a subset of harmful organic chemicals, mostly of anthropogenic origin, that are commonly classified as persistent organic pollutants (POPs) [1,2,3,4,5,6]. POPs are a class of carbon-based organic chemicals that are persistent, bioaccumulative and have long-range transport potential. There are three types of POPs present in the environment: (1) pesticides, especially organochlorine pesticides (OCPs) such as dichlorodiphenyltrichloroethane (DDT) and its metabolites; (2) industrial and technical chemicals including polychlorinated biphenyls (PCBs), polybrominated diphenyl ethers (PBDEs), and perfluorooctanesulfonate (PFOS); and (3) by-products of industrial processes including polychlorinated dibenzo-*p*-dioxins (PCDDs), polychlorinated dibenzofurans (PCDFs), and polyaromatic hydrocarbons (PAHs) [2]. PAHs do not strictly belong to POPs and they are only recognized as POPs under the Aarhus Protocol [7] because they can be efficiently metabolized and, therefore, prevent further bioaccumulation [8,9]. However, due to their lipophilicity and continuous release, PAHs are frequently classified as POPs in many studies [1,4,10,11,12]. Therefore, in this review, PAHs are discussed together with other POPs. Some commonly found POPs in food are listed in Table 1.

Most POPs are halogenated chemicals and the strong bond between carbon and chlorine/bromine/fluorine makes POPs resistant to the environmental degradation including chemical, biological, and photolytic reactions. For those non-halogenated POPs, their stable chemical structures also make them persistent in nature. POPs are resistant to environmental degradation including chemical, biological, and photolytic reactions. Therefore, once released, POPs can stay in the environment for a long time. Some POPs could have half-lives of years or decades and they can stay in the environment until they are taken up by plants and animals. POPs can bioaccumulate in the fatty tissue of living organisms and, therefore, become concentrated as they move through the food chain. POPs are a small subset of persistent, bioaccumulative and toxic chemicals (PBTs) that can travel great distance [35]. With the POPs group, some POPs are easily transportable, and others are not. POPs can move long distances in the atmosphere through air and water, even to places where they have never been used such as Antarctica and the Arctic area [36,37,38]. Due to these features, human and animals around the world could be exposed to POPs for extended long periods of time. Exposure to these pollutants, mostly at high levels, may cause various health problems such as endocrine disruption, cardiovascular diseases, cancers, diabetes, birth defects, and dysfunctional immune and reproductive systems [11,39,40]. The peak release of POPs was in the 1970s. Due to effective regulation and legislation, the current concentrations of many POPs are only one-tenth of the concentration at that time. In developed countries, many POPs are monitored to be below safety limits based on the known toxicology information. POPs are more of a threat to humans historically than now. However, POPs remain as a concern to human health because of the chronic exposure and the accumulation of POPs in the human body, especially in some developing countries.

Over 90% of human exposure to POPs is through the consumption of contaminated food, particularly food of animal origin [8,41,42]. Fishes are among the major sources of exposure to POPs [43,44,45]. To better protect public health, it is important to understand POP pathways into food, and the environment is among the significant pathways. POPs have been used and released to the environment through various human activities such as through the industrial and agriculture sector. The released POPs can easily contaminate crops, livestock, seafood and drinking water and pose a high risk to human health. For example, pesticides such as DDT and dieldrin have been widely used in the agriculture to increase crop yield and to kill unwanted pests in recent decades. However, the application of OCPs can easily introduce contaminants into water and crops and wildlife. Studies find that pesticide residue is among the most commonly found food contaminants [46]. Similarly, other POPs present in the environment such as PCBs, hexachlorobenzenes (HCBs), dioxin, and furans are also commonly found food contaminants.

To protect consumers from POP-contaminated food, many national and international agencies such as the European Food Safety Authority (EFSA), World Health Organization (WHO), US Environmental Protection Agency (EPA), and US Food and Drug Administration (FDA) [47] have developed regulations and guidelines to reduce the exposure to POPs. The Stockholm Convention requires its parties to take action to decrease the production, use and releases of the POPs on its list. The initial list was established in 2001 and included 12 POPs that are known as the “dirty dozen”. By 2019, an additional 17 POPs were added to the list. The Stockholm Convention is the most prominent, legal binding international framework that prevents the further accumulation of persistent toxic chemicals in food at the global level. As a result of the regulations and legal frameworks, the emission of dioxin and dioxin-like compounds have decreased considerably [48], as well as some hazardous pesticides in recent decades. Some monitoring programs have been employed to trace the trend of POPs such as the global monitoring plan for POPs by the United Nations Environment Programme (UNEP). The surveys show a consistent decline in PCDD, PCDF, PCB and DDT levels in human milk [49]. Overall, a general decrease was observed for POP levels in the environment and population in recent decades, demonstrating the effectiveness of the regulations and legal frameworks [49,50,51,52,53].

Currently, one challenging task in food safety is the assessment of health risks associated with POP dietary exposure. To assess the risk, information on the toxic effects of POPs and levels of dietary exposure is needed [54]. However, for most POPs, the toxic information is hardly available [54], mostly from quantitative structure activity relationship (QSAR) models [4,55], animal experiments [56,57], and epidemiological studies [51,58,59]. For POPs with quantified acute toxic effects, the estimation of their health risks is rather straightforward [54], and authorities could protect public health by limiting these POPs to below the safety limit. However, for most POPs found in food, the effects are long term and chronic. The continuous exposure and accumulation of POPs in the human body make it difficult to assess the dose–response relation between POPs and potential health problems [54], let alone determine the effective dose. For some POPs such as dioxin, furan, dioxin-like PCBs, PFOA and PFOS, some authorities have established the threshold values based on known toxicological information, but for a number of other commonly found POPs, the toxicological information is not clear, and the safety level is still unknown. Moreover, it is impossible to determine the threshold for genotoxic carcinogens such as some PAHs and their epoxides [60], because these POPs could damage DNA and there is no safety level of exposure [54]. For those POPs, a margin of exposure approach is used to determine the level of risks. The European Commission has established the maximum levels of certain PAHS on different foodstuff [61].

Dietary intake of POPs is usually estimated through the total diet study in various countries [51] and high performance analytical methods are needed in the study to determine POP levels in food. Currently, effective and rapid analytical approaches have been developed to determine trace amounts of POPs in food [62,63]. The approaches usually require multistep strategies including sample preparation and highly selective and sensitive instrumental techniques under strict quality assurance/quality control criteria [64]. The choice of sample preparation technique depends on the characteristics of the matrix. Sample preparation may include filtration, pH adjustment, extraction, clean-up and preconcentration procedures to ensure that the analytes are found at a suitable concentration level [5,65,66]. Popular sample extraction methods include Soxhlet extraction (SOX), solid–liquid extraction (SLE), solid-phase extraction (SPE), solid-phase microextraction (SPME), microwave-assisted extraction (MAE), liquid–liquid extraction (LLE), pressurized liquid extraction (PLE) and stir bar sorptive extraction (SBSE) [26,65,66,67,68].

Mass spectrometry (MS) has been considered among the most suitable instruments for the detection of POPs in food. MS has been widely applied in analyses of POPs in food due to its advantage of high sensitivity, selectivity and throughput. MS coupling with suitable separation techniques such as liquid chromatography (LC–MS), gas chromatography (GC–MS) and two-dimensional gas chromatography (GC×GC–MS) has been widely used for the detection of POPs in food due to its easy automation and high speed [69].

This article reviews the potential sources of POP contamination in food, the health impact of POPs, methods used to measure POPs in food and efforts to control POP contamination in food. To conclude, the article gives some conclusive remarks.

## 2. Sources of POP Contamination in Food

Food preparation usually involves multiple steps including processing, packaging, transportation and storage. Each step could be a potential source for POP invasion. Food could be contaminated by POPs through different paths. For example, raw materials may contain POPs that are transferred from the environment. Since POPs are resistant to degradation, they can stay in the environment for an extended period time. Previously released POPs in the environment are a major source of the POP contamination of food and feed supplies. Plant foliage uptake of POPs can effectively transfer POPs from air to plant and subsequently to food. Other sources of POPs are food preparation steps, during which POPs may artificially be introduced by humans. Table 2 summarizes food and POP contamination that has been detected.

### 2.1. OCPs

OCPs are among the main classes of POPs in the environment. They have been widely used in the agriculture industry since the second world war to protect plants. With a low cost, high toxicity and persistence nature, OCPs make the ideal candidates for treating soil and plants against various insects. However, OCPs could cause serious health problems for humans. The dietary exposure of OCPs is mainly from contaminated fatty foods including foods of animal origin such as eggs, dairy product and meat and foods of plant origin such as rice, fruit, vegetables and vegetable oil. Animals could be contaminated if they have access to OCP residues or OCP-contaminated feeds. Vegetables may be contaminated by root uptake from contaminated soil or from direct contact with OCPs.

Due to persistent toxicity, many OCPs such as aldrin, chlordane, DDT, dieldrin, endrin, heptachlor, HCB, mirex and toxaphene have been banned or restricted for use. However, these OCP residues remain in the environment due to their persistent nature. In addition, some OCPs are still used in areas such as Africa, South Asia, Central and South America. For example, high HCB levels were found in oysters from the pacific coast of Mexico [77]. A possible explanation for this is that the coast is bordered by extensive agriculture lands and HCBs were used in the Mexico’s agriculture until recently. Barber et al. confirmed the observation by reviewing the distribution and levels of HCB in the global environment [78]. The authors also pointed out that cod liver and cod liver oil may contain the highest accumulated HCB among fish caught for consumption and oil extraction. Another example is DDT, which was banned from agriculture use in most industrialized countries in the 1970s and 1980s because of its persistence and toxic effects [79]. However, many developing countries still use DDT in controlling the spread of malaria [80]. Since this public health benefit outweighs the potential adverse effects it may cause, the Stockholm Convention lists DDT in its annex B, allowing the production and use of DDT for the purpose of disease vector control. High concentrations of DDT were found in chickens and chicken eggs in malaria-controlled areas of South Africa where DDT was used as an indoor residual spray [81,82]. Levels of DDT and HCB were also detected in fish samples in India because of the elevated use of pesticides in agriculture until recently and the continuous use of DDT in controlling the spread of malaria [83].

### 2.2. PCB/BDE

PCB and PBDE are two POPs that have a similar structure and similar physicochemical properties. PCBs were widely used in the 20th century [84] as coolants and lubricants in transformers and capacitors and as hydraulic and heat exchange fluids in electrical apparatus [85]. PBDEs were extensively used from the beginning of the 1960s as flame retardants in electrical appliances, computers, TVs, insulation wires, cables and building materials [86]. The waste of electrical and electronic equipment has been released to the environment and resulted in high concentrations of PCBs and PBDEs in fatty foods such as meat, fish and poultry [87]. Humans could be exposed to PCB/BDE through inhalation, dermal contact and the consumption of contaminated food. Dietary exposure is the main route for PCB/BDE accumulation in the human body. For PCBs, there are 209 possible congeners and analysis is usually focused on toxic PCBs including dioxin-like PCBs (PCB-77, 81, 105, 114, 118, 123, 126, 156, 157, 167, 169, and 189) and indicator PCBs (PCB-28, 52, 101, 138, 153 and 180) [20,45,88,89,90]. For PBDEs, the most detected congeners in foodstuffs include 16 PBDEs (BDE-28, 47, 49, 53, 66, 85, 99, 100, 153, 154, 183, 196, 206, 207, 208 and 209) [45,91,92].

Although PCBs were banned in the 1970s, the old electrical transformers and building materials built before the ban still release PCBs into the environment and this is more significant than the incineration of PCB-containing waste and vehicle emission [93,94]. In addition, previously released PCBs persist in the environment and food sources [95]. Over 90% of the PCB body burden comes from food intake, mainly of fish, meat and dairy products. Son et al. studied the distribution of PCBs in 40 different foodstuffs in Korea and found that fish had approximately 4-fold higher concentrations of PCBs than rice [24]. However, even though foodstuff such as vegetables and rice have a low concentration of PCBs, they should be considered a potential and significant source of PCB intake because of the large amount consumed. Dietary makeup is a very important factor in the risk assessment of POP exposure. PBDE was banned in the USA and Europe in 2004. However, products manufactured with PBDE are still in use, which could be the major source of PBDE contamination. PCB/BDE-contaminated food can be found in countries where these chemicals have never been produced because of imported equipment containing PCB/BDE due to their long-range transportability.

### 2.3. Dioxin and Furan

PCDD and PCDF are known by-products of many industrial processes involving chlorinated chemical compounds. Humans are exposed to dioxins/furans mainly through high-fat food such as fish, meat and dairy products [96]. The waste incineration is a major source of dioxins and furans in developing countries where medical waste incineration and open burning are common [97]. Therefore, high levels of dioxins/furans were found in the milk and meat of animals in the vicinity of incineration plants [98]. For example, high levels of dioxins/furans and dioxin-like PCBs were found in some food groups (rice, vegetables, chicken, hen eggs, duck and crucian carp) from Zhejiang province, China, due to uncontrolled electronic recycling operations [90].

Accidents caused by artificially introduced POPs were well documented such as Yusho poisoning in Japan 1968 [99] and Yu-cheng poisoning in Taiwan in 1979 [100]. In the Yusho/Yucheng accidents, the victims were exposed to rice oil contaminated by extremely high concentrations of PCDFs/PCBs. The toxic equivalent (TEQ) intake for the accident victims was almost four-fold higher than for the general population [101]. The exposure continued over an extended period and caused serious health problems. On the other hand, for some other food poisoning accidents, such as the Belgian dioxin crisis in 1999 [102] and the 2008 dioxin accident in Ireland [103], the dioxin burden on the population did not increase and no consequences have been reported [103,104]. Because slow elimination leads to slow accumulation, the body burden depends more on long-term exposure than occasional exceedances of tolerable daily intake limits [105]. The Yusho and Yucheng accidents are different due to the extremely high levels of contaminants involved. There are other unintentional contaminated feed supply examples such as ball clays in poultry [106], kaolinic clay in Netherlands [107] and catfish feeds [17]. Accidental POP food and feed contamination is among the primary sources of single, extremely high dietary exposure to POPs. These incidents have aroused public attention to POP-related food safety issues. National and international agencies have developed policies to reduce consumer exposure to POPs by setting food and feed limits and effective surveillance programs to regulate POP levels in food and feed supply.

### 2.4. PAHs and Per- and Polyfluoroalkyl Substances (PFASs)

PAHs are a ubiquitous group of several hundreds of chemicals that comprise two or more benzene rings. PAHs are generated mostly because of the incomplete combustion of either natural sources such as coal and wood or man-made sources such as automobile emissions and cigarette smoke [9]. For non-smokers, diet exposure contributes over 70% of the human exposure [61]. PAHs may enter food through food preparation and handling methods. A dietary survey in the United Kingdom showed that cereals and oils/fats contributed a large part to the dietary intake of PAHs [108]. This typical PAH contamination usually occurs in technological processes such as direct fire drying, where food is exposed to combustion products [109]. Certain traditional home cooking methods such as grilling, roasting, frying and smoking could also result in high PAH concentrations in charcoal grilled/barbecued foods [110]. The packing materials and manufacturing are another source of potential food contamination [61].

Per- and polyfluoroalkyl substances (PFASs) and their derivatives are a group of chemicals comprising a fluorinated alkyl chain (generally C4–C18) and a hydrophilic functional group [111]. Since these chemicals are waterproof, oil-proof and heat resistant, they are used in a wide range of products and applications such as food packaging, non-stick cookware, cleaning agents, carpet, furniture, coating materials, etc. [112,113,114]. The most investigated PFASs include perfluorooctanesulfonate (PFOS) and perfluorooctanoic acid (PFOA). Perfluorooctanesulfonate is used in paper coating for food contact and perfluorooctanoic acid is a processing aid used in nonstick cookware [115]. PFOS has been detected in many food sources [116,117,118]. The main sources of PFOAs and PFOSs are fish, seafood, meat, eggs, dairy product and drinking water [112,114,119]. Recent studies on fluorinated chemicals in food packaging show their potentially significant contribution to dietary exposure to PFASs [119,120,121].

## 3. Health Effects 

The bioaccumulation of POPs in human fatty tissue and their persistent characteristics make POPs a major threat to human health. Exposure to these pollutants is associated with various serious health problems such as endocrine disruption, reproductive problems, cancer, cardiovascular disease, obesity and diabetes. Prenatal exposure to POPs not only poses adverse effects to the health of mothers but also to newborns. Recent studies have shown that prenatal exposure to POPs may be associated with a decrease in birth weight [122,123], child obesity, increased blood pressure [124] and endocrine-disrupting effects [125,126]. The toxic information is mainly obtained from animal test and epidemical studies. There are a lot of cofounding factors in the association of various health problems with POPs in food. Therefore, it is difficult to determine whether certain POPs are the cause of the associated health problems. Although the levels of POPs are decreasing, and POPs may not be a major threat to human health now compared to decades ago, the risk remains, and the safety margins still need to be improved. It is worth studying the possible health effects to better protect human health. In this section, various possible adverse health problems related to POPs are included but the determination of POPs as the causality of those adverse health problems needs further investigation. Possible human health hazards due to exposure to POP-contaminated food are given in Table 3.

### 3.1. Endocrine Disruption and Cancers

The endocrine system is responsible for regulating hormones that control many different body functions. During the last two to three decades, there is growing evidence that exposure to certain POPs likely causes endocrine disruption [131,132,133,134]. Endocrine-disrupting effects of POPs have the potential to cause adverse effects on the reproductive, neurological and immune systems, increasing the risk of the development of hormone-dependent cancers and affecting sexual differentiation, growth and development [134,135,136]. Some POPs in food have shown adverse effects such as cancers and hypospadias for fetal and infant males, while POP-contaminated food may result in breast cancer, cystic ovaries, and endometriosis for females [40].

Dioxin, furan and dioxin-like PCBs have been reported to influence the activation of transcription factors through binding with the aryl hydrocarbon receptor and interacting with hormone receptors, therefore, affecting normal hormone function [131,137]. These chemicals are known to be a dioxin-like family. They share a common mode of action and may cause the same adverse effects on humans [39]. A modified sex ratio at birth was found in the population of Seveso where individuals were accidentally exposed to 2,3,7,8-tetrachloro-dibenzo-*p*-dioxin (TCDD). TCDD is considered the most toxic compound in the dioxin-like family and is recognized as a human carcinogen by the International Agency for Research on Cancer (IARC). The epidemiological studies observed excess risks for all cancers in the population that was exposed to a 10–1000 times higher concentration of TCDD than the general population [39]. The cancer risk was also reported for people who consumed aquatic products, especially PCB-contaminated marine products [138].

OCPs (DDT, dieldrin, toxaphene, chlordane, mirex, endosulfan, HCB, etc.) are also considered endocrine disruptors [132]. There are a lot of studies suggesting human exposure to OCPs could lead to an increase in estrogen-dependent cancers [128,139,140]. The association of cancer with chlordecone exposure was identified in French West Indies where the population had been consuming chlordecone-contaminated food [128]. Higher levels of OCP residues were detected in breast cancer patients than in normal females despite their diet, age and geographical distribution [139]. An association of OCP with the risk of breast cancer was observed [140]. On the other hand, there are some studies that do not support the hypothesis that exposure to OCPs and PCBs would increase the risk of breast cancer [141,142,143,144]. Therefore, more studies are needed to investigate the possible relationship between OCPs and estrogen-dependent cancers.

Some PAHs such as benzo[a]anthracene and benzo[a]pyrene are classified as probable human carcinogens, whereas benzo[b]fluoranthene, benzo[j]fluoranthene, benzo[k]fluoranthene, and indeno [1,2,3-c,d]pyrene are classified as possible human carcinogens by the IARC [145]. Similarly, the US EPA classified seven PAHs including benzo[a]anthracene, benzo[a]pyrene, benzo[b]fluor-anthene, chrysene, benzo[k]fluoranthene, dibenzo[a,h]anthracene, and indeno[1,2,3-c,d]pyrene as probable human carcinogens (group B2) [146].

### 3.2. Cardiovascular and Metabolic Diseases

In addition to the deleterious effects of POPs on the endocrine system, there is evidence showing that exposure to POPs could lead to cardiovascular diseases [95,147] and metabolic diseases such as obesity and diabetes [127,148,149]. For example, in the 15 year period after the Seveso accident, an excess of cardiovascular mortality was noted in the population residing in the area contaminated by TCDD at that time [150]. The studies conducted in Sweden also showed that PCB and OCP exposure could be related to cardiovascular diseases [95]. The study conducted in the United States found that exposure to POPs may increase the risk of hypertension [151]. However, the conclusion is not definitive, and further studies are needed to confirm this observation. The study of serum POP levels in 428 adults from the Canary Islands did not show an association of hypertension risk with PCB and OCP levels in serum [58]. Conversely, the findings from that study suggested OCPs may induce divergent actions on blood pressure. Divergent effects on the risk of hypertension were observed, while PCBs were significantly associated with hypertension by analyzing a dataset obtained for 315 Inuit in 1992 [152].

Some recent studies found that elevated POP exposure had diabetogenic potential [127,153,154]. High levels of POPs in serum were found in diabetic and prediabetic individuals [148]. A significant association of dioxin-like compounds with the prevalence of metabolic syndrome such as high blood pressure, elevated triglycerides, and glucose intolerance was found among Japan’s general population [155]. Similarly, Yucheng women who had been previously exposed to PCBs and dioxins had an increased incidence of type 2 diabetes [129]. Exposure to DDT, dioxin and PFOA during pregnancy could lead to obesity in offspring [133]. A high risk of obesity was found in individuals who had high concentrations of OCP, PCB and PBDE in their body [149].

## 4. Detection Methods

Methods for the analysis of POPs in various food matrices have been developed in recent decades. The detection of POPs in food requires multistep strategies including sample preparation, highly selective and sensitive instrumental techniques, and quality assurance and quality control. The commonly used analytical methods for the detection of POPs in different food matrices are summarized in Table 4.

Because an extremely low detection limit is required for POP analysis, sample preparation is needed to reduce the matrix effect when analyzing foodstuff. Sample preparation used for the detection of POPs in food involves multiple steps including filtration, pH adjustment, extraction, clean-up and enrichment procedures to ensure that the analytes are detected at a suitable concentration level [5,65,66]. Different sample preparation techniques have been developed, including supercritical fluid extraction, solid-phase extraction, solid-phase microextraction, microwave-assisted extraction, liquid–liquid extraction, liquid-phase microextraction, pressurized liquid extraction and stir bar sorptive extraction [3,67,156,157,158].

### 4.1. Extraction

The selection of sample preparation techniques is dependent on the characteristics of the sample matrix. For liquid/aqueous samples, liquid–liquid extraction (LLE) is the most conventional extraction method. The application of LLE has been widely accepted in standard methods for the analysis of POPs in water and milk including PCDD/Fs and PCBs and OCPs [29,173]. LLE separates compounds based on their relative solubilities in two immiscible liquids, usually water and organic solvent. Therefore, it requires large amounts of organic solvent [156]. To reduce solvent consumption, a modified LLE method, dispersive liquid–liquid microextraction, was developed [67,174]. Dispersive liquid–liquid microextraction is mainly used to extract organic analytes (PCB, PBDEs, OCPs, and PAHs) from water samples [175]. Due to its simplicity and low cost, dispersive liquid–liquid microextraction becomes very popular in separation science.

Solid-phase extraction is another alternative solvent reduction method for liquid samples. It has been used by the US EPA as an alternative to LLE when analyzing organic compounds in water and wastewater [176,177]. Solid-phase extraction separates compounds in a liquid mixture based on the different affinities for a solid phase (sorbent) between an analyte and interferences. It has been used to extract PFOS, PFOA from water samples [26]. Compared to the traditional LLE, solid-phase extraction could greatly reduce solvent consumption and its operation is quite simple and inexpensive. However, conventional solid-phase extraction has some limitations such as potential analyte loss during the preconcentration step and clogging of the sorbent beds [156,177]. Recently, solid-phase microextraction and stir bar sorptive extraction have been developed to reduce sample loss and contamination. The limitation of the stir bar sorptive extraction is the restriction on polar compounds, while solid-phase microextraction needs a clean-up process after sample extraction. Both methods have been used to extract OCPs, PCBs and PBDEs in water samples [158,178,179].

Common extraction techniques for solid samples include Soxhlet extraction, pressurized liquid extraction, supercritical fluid extraction, microwave-assisted extraction, ultrasonic-assisted extraction and matrix solid-phase dispersion extraction. The conventional Soxhlet extraction is still among the most common methods used for various matrices and analytes, especially for dioxins/furans and dioxin-like PCBs in food [26,29]. However, this extraction process is very time-consuming and uses large amounts of solvents. In addition, the need for evaporation after sample extraction excludes the application of thermolabile compounds which may degrade due to the prolonged heating.

To overcome the limitations of conventional extraction methods, alternative extraction methods such as pressurized liquid extraction [180,181], microwave-assisted extraction [157], ultrasound-assisted extraction [76] and supercritical fluid extraction [159,182] have been developed. In pressurized liquid extraction, elevated temperature and pressure are used to achieve high extraction of the components from sample matrices. Pressurized liquid extraction has been widely used for the extraction of PAHs, PCBs, PCDFs, PCDDs in fatty foods including egg, fish and meat samples [183]. Microwave-assisted extraction uses microwaves to heat the solvent and increase the solvent penetration into the sample matrix. It has been widely used to extract OCPs from food [29]. Microwave-assisted extraction is very attractive because it requires less extraction time, reduces solvent usage, and improves extraction yield. However, it has some drawbacks such as expensive equipment, a polar solvent and a clean-up process afterwards. Ultrasonic-assisted extraction is a simple and inexpensive alternative that can reduce operating temperature with the help of ultrasound waves. Ultrasonic-assisted extraction has been applied for the detection of OCPs and PAHs in food [76]. Supercritical fluid extraction uses supercritical fluid as the extracting solvent to separate analytes from the mixture. Carbon dioxide has been the mostly used supercritical fluid. This method has been used to extract OCPs from egg, butter, oil and meat products [159,160,184,185]. The common drawback of these extraction methods is that they all need cleanup steps afterwards because some interfering organic compounds are extracted together with POPs. The cleanup process removes the interfering substances to prepare the extract for instrument analysis. Automated clean-up systems have been developed as manual clean-up processes are tedious. However, because of the high investment needed for automated clean-up systems, manual clean-up is still an attractive method [18]. LeDoux reviewed some commonly used clean-up methods used for the detection of POPs in food such as freezing centrifugation, liquid–liquid portioning, gel permeation chromatography, solid-phase extraction, solid-phase microextraction and concentrated sulfuric acid [185].

### 4.2. Separation and Detection

Mass spectrometry (MS) coupled with chromatography is the mostly widely applied technique for POP quantification in food and environmental matrices [20,89,186,187,188,189,190]. Since food-based matrices are complex, selectivity is the primary concern. An overview on different separation and detection techniques is presented in Table 4.

Gas chromatography (GC) is the most commonly used technique for separation. The GC separation is dependent on the boiling points of the compounds and their interactions with the stationary phase of the column. Most POPs are semi-volatile, and their polarities are between moderate and non-polar. These physicochemical properties make most POPs well suited to being measured by GC–MS, except for PFAS-related chemicals, which are always measured using LC–MS/MS method. However, there is no single column that can separate all congeners of PCBs and dioxins/furans. To overcome this difficulty, comprehensive two-dimensional GC was introduced. When passing two columns, there are two degrees of separation based on different physiochemical properties. Compared to single columns, the two-dimensional GC can significantly improve selectivity (peak capacity) and sensitivity.

The majority of the analytical methods for the detection of POPs is GC coupled with conventional detectors such as an electron capture detector [191,192] and MS operated in different ionization modes (negative chemical ionization and electron ionization) [191]. The electron capture detector is a low-cost detector and is mostly used in the analysis of PCBs and OCPs in different foodstuff [29,183]. However, recently, MS has become the most commonly used detector for POP analysis. Several techniques are used in MS to generate ions. One widely used technique for the detection of POPs in food is GC coupled with MS in the selected ion mode. The selected ion mode could improve selectivity by focusing on a selected number of relevant masses corresponding to analytes. However, significant fragmentation in the electron ionization condition could affect selectivity in some samples [20]. Electron capture negative chemical ionization is an alternative softer ionization and is very useful for the detection of POPs. PCB and PBDEs can be analyzed using GC–MS in either the electron capture negative ionization mode or the electron ionization mode. For liquid chromatography, atmospheric ionization such as electrospray and atmospheric pressure chemical ionization are two widely used techniques for the detection of POPs in food. Atmospheric pressure chemical ionization was also coupled with GC to analyze dioxins and PBDEs [188].

GC coupled with ^13^C-labeled isotope dilution high-resolution mass spectrometry (HRMS) is considered a standard method for the detection of specific POPs such as dioxins and furans [16,19,193]. The direct ^13^C-labeled isotope dilution provides reliable quantification. However, due to the high cost of the equipment and the need for skilled technicians for ^13^C-labeled isotope dilution HRMS, other MS instruments such as time-of-flight mass spectrometry (TOF–MS) have been utilized. TOF–MS is promising when coupled with suitable GC methods such as comprehensive two-dimensional gas chromatography. GC×GC TOF–MS has been successfully applied to detect dioxins and PCBs in food [14]. The specificity of GC×GC TOF–MS could be improved by either operating the instrument in tandem model (MS/MS) or improving the chromatographic separation [14]. Recent studies have shown that GC coupled with triple-quadrupole tandem MS had a high performance similar to GC-HRMS for the detection of POPs in food and feedstuff samples such as vegetable oil and fish [19,107]. Atmospheric pressure gas chromatography (APGC) triple quadrupole has also proved to have sufficient sensitivity and selectivity in the analysis of dioxins and PCBs in food and feed samples [89].

### 4.3. Quality Control and Assurance

For quality control and assurance, quality control (QC) samples are needed to ensure the reliability and comparability of POP quantification data. QC samples include blank, spike and reference samples. A blank sample is an analyte-free matrix. It is treated identically to target samples extracted in the same batch, including exposure to all glassware, equipment, solvents, reagents, and internal standards [194]. A blank sample is used to demonstrate that the system (laboratory, reagents, glassware, etc.) is free of contaminants and to ensure that the extraction process is under control. For food sample analyses, corn oil is often used as a blank sample [67]. Matrix-spiked samples are made by spiking individual analytes prior to their extraction. The recovery of the spiked analytes is used to check the accuracy of the analytical method. Reference material is used as a quality control to ensure that the analytical method is accurate. Many reference materials are provided by national authorities and commercial institutes such as the National Institute of Standard and Technology [67]. In addition to QC samples, the quality of analytical data needs to be assessed through interlaboratory comparison studies. International monitoring programs provide interlaboratory comparisons to check the performance of different laboratories.

## 5. Efforts to Control POP Food Contamination

To protect human health from POPs, it is crucial to develop control methods to prevent POPs from entering food. There are two kinds of efforts: reducing POPs in the environment and preventing POPs from entering the food chain.

### 5.1. Monitoring of POPs in Food

Due to the persistence and long-range transportability of POPs, regulations at both national and international levels have been issued to protect the environment and public health. A comprehensive overview on the various national and international regulatory frameworks on PBTs and POPs was published by Abelkop et al. [195] and Matthies et al. [196]. International treaties on managing POPs and other hazardous chemicals include the Basel Convention on the Control of Transboundary Movements of Hazardous Wastes and their disposal, the Rotterdam Convention on the Prior Informed Consent Procedure for Certain Hazardous Chemicals and Pesticides in International Trade and the Stockholm Convention on prohibiting or restricting the use and production of certain POPs [195,196]. The Stockholm Convention is among the most important guides in regulating POPs at the global level. It entered in force in 2001. In total, 179 countries adopted this convention until 2014. To implement the Stockholm Convention, all participating countries developed their own implementation plans to eliminate or reduce the release of POPs from intentional and unintentional productions. All POPs in the Stockholm Convention list have been banned for use in Europe, North America and many South American countries. There are also some regional regulations with a smaller jurisdiction size [195,196] such as the Regulation for Registration, Evaluation, Authorization and Restriction of Chemicals (REACH, EC no. 1907/2006 as amended) [197], EU regulation for the Placing of Plant Protection Products on the Market (EC 1107/2009) [198], the United Nations Economic Commission for Europe (UNECE) POPs Protocol [7] and the North American Sound Management of Chemicals [199]. Some national regulations [196] are also in place such as the chemical management plan under the Canadian Environmental Protection Act [200] or the U.S. Toxic Substances Control Act [201]. There are other national legislations and regulations in countries such as India [202] and China [203]. These national and international regulatory efforts resulted in a time-related decrease in POPs [204]. However, many developing countries are still using some banned POPs for agriculture and public health purposes because of their poor economy, which is allowed by the Stockholm Convention because this public health benefit outweighs the potential adverse effects it may cause [11]. POPs remain a serious threat to human health. Global effort and resources are needed to reduce and ultimately eliminate the release of POPs.

In addition to these regulations, to ensure food safety, international bodies developed surveillance programs to monitor POP levels in food to protect the public from contaminated food. Since dioxin-like compounds are among the most toxic chemicals, many government and non-governmental organizations established the safety limit for tolerable intake of PCDD/Fs and PCBs [205,206,207]. For example, the European Union set the maximum limits for PCDD/Fs and dioxin-like PCBs in food products in its regulation since 2001 (the most recent one is EC regulation 1259/2011 [208]). There are also similar regulations focusing on some PAHs and PFOAs [9]. Both national and international monitoring programs have been established to make certain that POP contamination in food is below the harmful level. The regulations on feed supplies are also needed to prevent POPs from entering the food chain. A POP-contaminated feed supply could significantly contaminate the animal food supply such as in the Belgian dioxin crisis in 1999 [102] and in the Ireland dioxin accident in 2008 [103]. However, it is important to understand that such programs can only reduce the risk of POP contamination in food and they cannot completely eliminate the supply of POP-contaminated food to consumers [71]. The Yusho and Yu-cheng accidents demonstrate the need to prevent accidental or deliberate contamination. However, to better protect public health from POP contamination, strong legislation enforcements are needed to minimize the continuous release and ensure the rational use of pesticide and medicines.

### 5.2. Removal Methods

Another approach to reduce the possibility of POP contamination in food is to develop methods to remove POPs present in the environment and subsequently reduce the risk of food contamination. Traditional technologies include incineration, solvent extraction, gas-phase chemical reduction, alkali metal reduction and landfilling [1]. However, these traditional methods have proven to be insufficient in completely removing POPs. Moreover, these methods are expensive and may produce more toxic compounds during degradation [2,71,209]. Bioremediation is an alternative method that uses microorganisms to biodegrade pollutants in an environmentally friendly manner. Approaches to bioremediate POPs have been reviewed [2,12].

### 5.3. Dietary Make-Up

Dietary make-up also affects individual exposure to POPs. Due to the fat solubility of POPs, high-fat products such as milk, animal food and its products are easier to contaminate by POPs than other products. Dietary strategies to reduce POP exposure include decreasing the consumption of meat, dairy and fish or selecting the lowest fat option [206]. Other POP contamination pathways such as food packaging and cooking processes also need to be considered. Exposure to contaminants from processed foods can be reduced by using safer storage methods such as edible films and coatings [210] and processing methods such as the use of indirect heat and purified oil [71,211].

## 6. Concluding Remarks

Despite the declining trend of POPs in the environment due to national and international control actions [30,204], POPs are still a global concern. Some developing countries still use banned POPs due to their poor economy [11]. Now national regulations and legislations play an important role in reducing the use of POPs but regulation at the international level is needed to enforce countries to minimize the continuous release and ensure the rational use of pesticide and medicines. Therefore, strong regulations at the global level are needed reduce the use of POPs and ultimately eliminate their release. Second, the chemicals under regulation are only a small fraction of widely used commerce chemicals. The Stockholm Convention [6] defines criteria for new POP candidates in terms of their persistence, long-range transport, bioaccumulation and toxicity. Recently, some studies have shown many potential chemicals that may pose a high risk to human health and that need to be monitored [212]. More studies are needed to investigate these emerging pollutants in food matrices and to develop new analytical methods for the detection of the emerging POPs in food. Since the determination of new POPs in food matrices has not been extensively investigated, sample preparation techniques specific to complex matrices are needed to provide the accurate measurement of emerging pollutants [65].

Regarding the health effect of POPs on humans, further studies are needed to understand their toxicological mechanisms. Toxicological information is needed to assess the risks of POP dietary exposure. The fundamental mechanism in which POPs bind to the aryl hydrocarbon receptor appear unlikely to predict the risk of some diseases [213]. Regulatory limits are set based on comprehensive toxicological knowledge and they are designed to ensure that human or animal exposure is at the safety level. For PCDD/F and dioxin-like PCBs, the tolerable daily intake was set by the WHO in 2000 as 1–4 pg TEQ kg^−1^ body weight [214]; the tolerable weekly intake was recommended by the European Food Safety Authority (EFSA) in 2018 as 2 pg WHO-TEQ kg^−1^ body weight [215]; and the provisional tolerable monthly intake was set by the Joint Expert Committee on Food Additives (JECFA) of the Food Agriculture Organization (FAO) and the WHO in 2001 as 70 pg WHO-TEQ kg^−1^ body weight [216]. The European Union has established the maximum levels for PCDD/Fs and the sum of PCDD/Fs and dioxin-like PCBs in various foodstuffs in 2011 [208,217]. For example, for the sum of PCDD/Fs and dioxin-like PCBs, the EU maximum level is 4 pg TEQ/g fat for beef, 5.5 pg TEQ/g fat for milk, 5.0 pg TEQ/g fat for egg, 6.5 pg TEQ/g wet weight for fish except eel and 1.5 pg TEQ/g for compound feed [208]. For DDTs and HCHs, the FAO and the WHO set up acceptable daily intakes to be less than 10,000 ng/kg body weight/day and 5000 ng/kg body weight/day, respectively [218]. For PAHs, the regulatory maximum levels on traded food is 2.0 µg/kg for BaP and 12.0 µg/kg for PAH4 in smoked meat and smoked meat products [219]. Currently, only some POPs have the safety limit established based on their known toxic effects, and a lot of new emerging POPs have no determined threshold. Therefore, to better protect human health, more studies should focus on understanding the toxic effects of emerging POPs. In general, more detailed and systematic research is required to predict the risk of POPs and to improve POP regulations, with a better understanding of toxic effects and advanced analysis methods.

More effective and environmentally friendly techniques are also needed to remove POPs from the environment. Although some techniques are in the experimental and pilot stages, the research concerning POP removal is far from enough and further investigation is needed to develop new environmentally friendly and human-friendly techniques [1].

## Figures and Tables

**Table 1 ijerph-16-04361-t001:** Commonly found persistent organic pollutants (POPs) in food.

POPs Class	POPs	Structure	Reference
Chlorodibenzo-p- dioxin (CDD)	2,3,7,8-tetraCDD; 1,2,3,7,8-pentaCDD; 1,2,3,4,7,8-hexaCDD; 1,2,3,6,7,8-hexaCDD; 1,2,3,7,8,9-hexaCDD; 1,2,3,4,6,7,8-heptaCDD; octaCDD	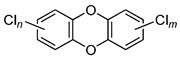	[13,14,15,16,17,18,19,20]
Chlorodibenzo furan (CDF)	2,3,7,8-tetraCDF; 1,2,3,7,8-pentaCDF; 2,3,4,7,8-pentaCDF; 1,2,3,4,7,8-hexaCDF; 1,2,3,6,7,8-hexaCDF; 2,3,4,6,7,8-hexaCDF; 1,2,3,7,8,9-hexaCDF; 1,2,3,4,6,7,8-heptaCDF; 1,2,3,4,7,8,9-heptaCDF; octaCDF	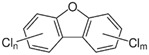	[13,14,16,17,18,19,20]
Polychlorinated biphenyls (PCBs)	PCB-28; PCB-52; PCB-70; PCB-77; PCB-81; PCB-101; PCB-105; PCB-114; PCB-118; PCB-123; PCB-126; PCB-138; PCB-153; PCB-156; PCB-157; PCB-167; PCB-169; PCB-170; PCB-180; PCB-189	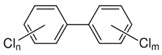	[13,15,16,18,19,20,21,22,23,24]
Polybrominated diphenyl ethers (PBDEs)	pentaBDE; decaBDE; heptaBDE	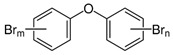	[22,23]
Hexabromocyclododecanes (HBCDs)	α-HBCDβ-HBCDγ-HBCD	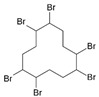	[25]
Hexabromobiphenyl	hexabromobiphenyl	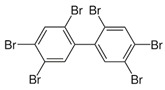	[26]
Hexachlorobutadiene (HCBD)	HCBD	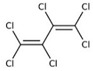	[27]
Polychlorinated naphthalenes (PCNs)	PCN	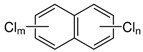	[28]
Short-chain chlorinated paraffins (SCCPs)	SCCPs(C_10–13_)	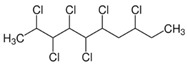	[27]
Organochlorine pesticide (OCPs)	*p,p′*-dichlorodiphenyltrichloroethane (DDT)	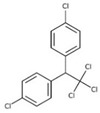	[8,21,29,30,31,32]
*o, p′*-DDT	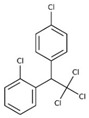
*p, p′*-dichlorodiphenyldichloroethylene	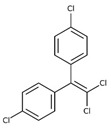
*p, p′*-dichlorodiphenyldichloroethane	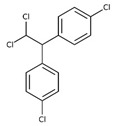
*o,p′*-dichlorodiphenyldichloroethane	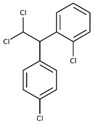
*o,p′*-dichlorodiphenyldichloroethylene	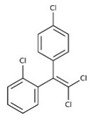
*p,p′*-1-chloro-2,2-(bis-(4-chlorophenyl)ethylene	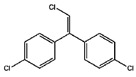
*p,p′*-1-chloro-2,2-bis(p-chlorophenyl)ethane	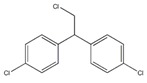
*cis-*chlordane(α-chlordane); *trans*-chlordane(γ-chlordane)	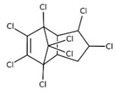
*cis* nonachlor; *trans*-nonachlor	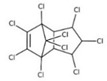
oxychlordane	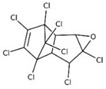
heptachlor	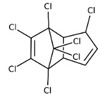
aldrin	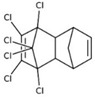
α-endosulfan; β-endosulfan	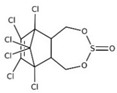
endosulfan sulfate	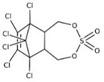
α-hexachlorocyclohexane (α-HCH); β-hexachlorocyclohexane (β-HCH); γ-hexachlorocyclohexane (lindane)	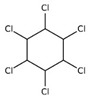
Hexachlorobenzene (HCB)	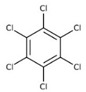
dieldrin;endrin	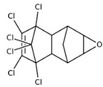
mirex	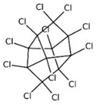
chlordecone	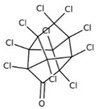
pentachlorophenol (PCP)	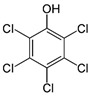
pentachlorobenzene (PeCB)	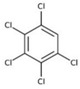
toxaphene	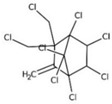
Perfluorinated compounds	perfluorooctanesulfonate (PFOS)	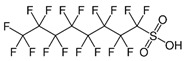	[33]
Perfluorooctanoic acid (PFOA)	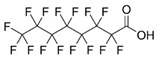
Polyaromatic hydrocarbons (PAHs)	anthracene	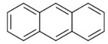	[34]
pyrene	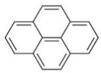
benzo(a)anthracene	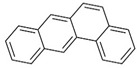
benzo(k)fluoranthene	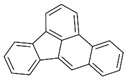
benzo(a)pyrene	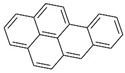
indeno(1,2,3cd)pyrene	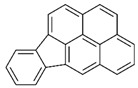
dibenzo(a,h)anthracene	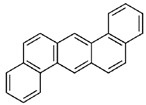
benzo(g,h,i)perylene	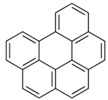

**Table 2 ijerph-16-04361-t002:** Summary of food contaminated with POPs.

Foodstuff	POPs	Reference
Egg	Dioxins/furans, PCBs, OCPs, PFCs and HBCDs	[15,20,23,28,29,52,70]
Dairy product (milk, butter, cheese, cream, yogurt, ice cream, etc.)	Dioxins/furans, PCBs, OCPs and PAHs	[13,15,20,29,42,71,72]
Meat and meat product (pork, chicken, beef, sausage, etc.)	Dioxins/furans, PCBs, OCPs, HCBD and PCN	[15,20,28,31,32,52,70]
Grain, flour and bran	PAHs	[71]
Rice, Fruit and vegetable (cabbage, carrot, potato, etc.)	OCPs, PCBs and PAHs	[3,42,73,74]
Honey	OCPs	[71]
Oil (vegetable oil, olive oil, etc.)	Dioxins/furans, PCBs, OCPs and HBCDs	[18,31,52]
Fish	OCPs, PCBs, PBDEs, PFOS, Dioxins/furans and HBCDs	[20,21,22,23,25,26,27,28,29,59,72,75]
Mussel	OCPs, PCBs and PBDEs	[26,30,76]
Oyster	PAHs	[71]
Water	PFOS, OCPs, PCBs and PAHs	[3,26]

**Table 3 ijerph-16-04361-t003:** Health hazards associated with POPs in food.

POP	Possible Hazards	Reference
PAHs	Mutagenicity and carcinogenicity, DNA damage, oxidative stress, impaired male fertility, respiratory diseases, cognitive dysfunction among children and cancer (breast cancer)	[11,28,71,88]
OCPs	Neurological symptoms, endocrine disruption, infertility and fetal malformation, diabetes, cancer (breast cancer, testicular, prostate and kidney cancer), reproductive problems, cardiovascular problems, high blood pressure, glucose intolerance and obesity	[11,28,40,127,128]
Dioxins/furans	Language delay, disturbances in mental and motor development, cancer, diabetes, endocrine disruption, high blood pressure, glucose intolerance and cardiovascular problems	[11,28,129]
PCBs	Endocrine disruption, neurological disorders, liver injury, diabetes, cancer (breast, prostate, testicular, kidney, ovarian and uterine cancers), cardiovascular problems and obesity	[11,28,129]
PBDE	Reproductive problems, cancer(testicular), diabetes, obesity and cardiovascular problems	[11]
PFOS and PFOA	Breast cancer	[11]
HBCD	Endocrine disruption, reproductive problems and behavioral effects	[130]
PCN	Cancers	[28]
PCDE	Cancers	[28]

**Table 4 ijerph-16-04361-t004:** Analytical methods for the detection of POPs in food.

Type	Method	Description	Reference
Extraction	Soxhlet extraction (SOX)	Suitable for solid samples; efficient but time consuming and possible low analyte recovery	[18,22,26,29]
	Solid–liquid extraction (SLE)	Suitable for solid samples; expensive and uses large volumes of organic solvents	[29]
	Pressurized liquid extraction (PLE)	Suitable for solid samples; highly automated but need expensive specialized equipment	[29]
	Supercritical fluid extraction (SFE)	Suitable for solid matrices; high efficiency, selectivity and low solvent volume, but need clean-up step	[29,159,160]
	Microwave-assisted extraction (MAE)	Suitable for solid samples; high efficiency but need clean-up step	[29,157,161,162,163]
	Ultrasonic-assisted extraction (UAE)	Suitable for solid samples; require low solvent volumes but need to optimize different operating factors	[70]
	Matrix solid-phase dispersion (MSPD)	Suitable for solid, semi-solid and viscous sample matrices; combines extraction and cleanup within a single step but need trials and errors to pick the right sorbent	[164]
	Liquid–liquid extraction (LLE)	Suitable for liquid/aqueous sample; high efficiency and selectivity but tedious and requires large amounts of organic solvents	[29]
	Solid-phase extraction (SPE)	Suitable for aqueous/liquid samples; requires large sample volumes	[26,165]
	Stir bar sorptive extraction (SBSE)	Suitable for liquid/aqueous samples; simple and solvent-less, but not suitable for polar compounds	[65,156]
	Solid-phase microextraction (SPME)	Suitable for liquid/aqueous samples; simple, solvent-less, less sample loss and contamination, but may need a clean-up process	[26,166]
Separation	Gas chromatography (GC)	Good separation potential but restricted to use on more volatile compounds, e.g., high-resolution gas chromatography (HRGC), Atmospheric Pressure Gas Chromatography (APGC)	[3,18,59,167]
	Liquid chromatography (LC)	Good for polar water-soluble class of chemicals; poor separation potential, e.g., High-Pressure Liquid Chromatography (HPLC)	[4,168]
	GC×GC	Good separation potential but restricted on more volatile compounds	[14,164]
Detection	Electron capture detector	Most commonly used detection method with low detection limits	[29,168,169]
	Mass spectrometry (MS) in the negative chemical ionization mode	Better sensitivity but restricted on non-polar POPs	[169]
	MS in the electron ionization mode	Better sensitivity and selectivity due to abundant fragmentation but restricted on non-polar POPs	[18,170]
	MS in the selected ion monitoring mode	Better sensitivity but the selected ion window may need to be monitored	[32,171]
	High-resolution mass spectrometry (HRMS)	High sensitivity but expensive	[14,18,59]
	MS/MS	Improves sensitivity and selectivity compared to single quadrupole MS, e.g., ion trap MS/MS; triple quadrupole MS/MS	[14,88,107,164,172]
	Time-of-flight TOF–MS	Wide mass analysis range but poor instrument limits of detection	[14,47]

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
