# Peer review of "Persistent Organic Pollutants in Food: Contamination Sources, Health Effects and Detection Methods"

_ijerph, 2019, doi:10.3390/ijerph16224361_

Round 1

Reviewer 1 Report

the manuscript was very much improved and can be accepted for publication. Only a few corrections should be made:

l. 55 Delete 'soil'.

l. 100 Information 'is', not 'are'.

l. 267 Replace 'working mechanism' with 'mode of action'.

l. 458 Replace both 'etc.' with 'et al.'

l. 534 Replace 'United Nations Environment Program' with 'Stockholm Convention [6]'.

Author Response

We appreciate the reviewer for the constructive comments. We carefully read the comments and revised our manuscript accordingly. The reviewer’s comments dramatically improved our manuscript. Below we give the point-by-point responses to the reviewer’s comments. We hope our responses satisfy the reviewer.

55 Delete 'soil'.

[Author reply] Thanks for your suggestion. We deleted ‘soil’ from the manuscript.

100 Information 'is', not 'are'.

[Author reply] Thanks for catching this. We changed “are” to “is”.

267 Replace 'working mechanism' with 'mode of action'.

[Author reply] Thanks for the suggestion. We replaced “working mechanism” with “mode of action”

458 Replace both 'etc.' with 'et al.'

[Author reply] Thanks for catching this. We replaced ‘etc.’ with ‘et al.’

534 Replace 'United Nations Environment Program' with 'Stockholm Convention [6]'.

[Author reply] Thanks, we replaced 'United Nations Environment Program' with 'Stockholm Convention [6]'

Reviewer 2 Report

This manuscript has been improved in many details, but the structural difficulties remain. POPs include so many different and toxicologically very mixed groups of compounds, some are extremely toxic, many are not. Therefore an attempt to describe all of them is bound to be either too general or only a listing of everything that somebody has suggested. This text may be useful for somebody in collecting literature on the topic, but it is not very balanced source as to toxicity or risks of those compounds. It looks like the authors are more chemistry oriented than toxicology oriented, because many central principles of toxicology are vague. Language needs attention, especially singular/plural checks (often the subject noun is singular, but verb is plural).

Some examples of the logical inconsistencies:

Line 47: What do the authors mean by a small subset of chemicals that have long-range transportability? The problem is that within the POP group there are chemicals that are easily transportable, but others are not. So it would be reasonable to formulate the expression in a different manner.

Lines 60-64: The text implies that POPs can easily contaminate food and pose a high risk to human health. Sure they can, but do they in real life? The uncontrolled use of POPs (notably dioxins and PCBs and most chlorinated pesticides) took place almost half a century back, peak releases to the environment were in 1970s. That residues can still be found using highly specialized analytics, does not tell much of high risk to human health. This separation remains unclear still troughout the MS, although there has been improvement. I recommend that the authors have still another look at the MS in this sense.

Line 19 illustrates the same problem, it is a nonrelevant issue, how many POP residues can be found. It rather indicates analytical capabilities. In food analysed using high-performance GC-MS usually most of the 17 toxic PCDD/Fs can be found, and many more PCB congeners.

Line 131: high toxicity to insects or to people? High toxicity towards pest is the aim, but toxicity to humans is very variable among the group.

Line 197: Ref. 97 describes increases of concentrations in people losing weight (which can be expected for compounds stored in fat), so the ref. does not deal with low-dose chronic exposure being more serious than high-dose short-term exposure. It looks like there is a misunderstanding on the kinetic behaviour of slowly eliminating POPs. Because body burden is much more than a daily dose in compounds with a long half-life (e.g. with TCDD body burden at constant intake is about 5000 times the daily dose), it means that a short-term exposure to doses exceeding safety margins set by authorities, do not necessarily increase the body burden much. This was clearly seen in Belgian PCB/dioxin incident as correctly referred to. A correct formulation could be something like: Because slow elimination leads to slow accumulation, the body burden depends more on long-term exposure than occasional exceedances of tolerable daily intake limits. A possible recent reference is Tuomisto et al, A pharmacokinetic analysis … in Toxicology Letters 2016:261:41-48.

Line 225: Again, the big threats are more historical than a contemporary problem,  although some risks remain and the safety margins are not great.

Line 246: The authors probably mean aryl hydrocarbon receptor (AHR). The wording is a bit unclear, do the authors mean that dioxins bind to hormone receptors which is probably not true, but the receptors interact.

Line 279-280: There is a caveat in hot spot studies, because as stated earlier in this MS, dioxin intake is mainly from food, and very little food comes today from local sources, especially very little of animal source food and fish. So it would be very hard to explain why a local hazardous waste site could be crucial, at least there should be thorough analytical evidence.

Author Response

We appreciate the reviewer for the constructive comments. We carefully read the comments and revised our manuscript accordingly. The reviewer’s comments dramatically improved our manuscript. Below we give the point-by-point responses to the reviewer’s comments. We hope our responses satisfy the reviewer.

This manuscript has been improved in many details, but the structural difficulties remain. POPs include so many different and toxicologically very mixed groups of compounds, some are extremely toxic, many are not. Therefore an attempt to describe all of them is bound to be either too general or only a listing of everything that somebody has suggested. This text may be useful for somebody in collecting literature on the topic, but it is not very balanced source as to toxicity or risks of those compounds. It looks like the authors are more chemistry oriented than toxicology oriented, because many central principles of toxicology are vague. Language needs attention, especially singular/plural checks (often the subject noun is singular, but verb is plural).

[Author reply] Thanks for your suggestion. Your comments and suggestions are very useful for us to improve the quality of our manuscript. Yes, we totally agree that POPs include a lot of chemicals that have very different toxic properties. We made a hard decision trying to cover most POPs commonly found in foods. To differentiate them, in the manuscript, we discussed different POPs separately. For example, Dioxin/Furan, PCB, PBDE are discussed in different sessions. Since most of our teams have the background in Chemistry, we tried our best to cover the toxicology aspects of POPs and revised the manuscript accordingly.

Line 47: What do the authors mean by a small subset of chemicals that have long-range transportability? The problem is that within the POP group there are chemicals that are easily transportable, but others are not. So it would be reasonable to formulate the expression in a different manner.

[Author reply] Thanks for the comment. Accordingly, we change the sentence to “POPs are a small subset of persistent, bioaccumulative and toxic chemicals (PBTs) that can travel great distance [35]. Within the POPs group, some POPs are easily transportable, and others are not.”

Lines 60-64: The text implies that POPs can easily contaminate food and pose a high risk to human health. Sure they can, but do they in real life? The uncontrolled use of POPs (notably dioxins and PCBs and most chlorinated pesticides) took place almost half a century back, peak releases to the environment were in 1970s. That residues can still be found using highly specialized analytics, does not tell much of high risk to human health. This separation remains unclear still troughout the MS, although there has been improvement. I recommend that the authors have still another look at the MS in this sense.

[Author reply] Thanks for the comment. We added the following sentences in the manuscript. “The peak releasement of POPs was back in 1970s. Due to the effective regulation and legislation, the current concentrations of many POPs are only one tenth of the concentration at that time. In developed countries, many POPs are monitored to be below safety limits based on the known toxicology information. POPs are more of a threat to human historically than now. However, POPs remain as a concern to human health because of the chronic exposure and the accumulation of POPs in human body, especially in some developing countries.”                                                                                                                                                                                          

Line 19 illustrates the same problem, it is a nonrelevant issue, how many POP residues can be found. It rather indicates analytical capabilities. In food analysed using high-performance GC-MS usually most of the 17 toxic PCDD/Fs can be found, and many more PCB congeners.

[Author reply] Thanks for the comment. To avoid confusion, we deleted this sentence from the manuscript.

Line 131: high toxicity to insects or to people? High toxicity towards pest is the aim, but toxicity to humans is very variable among the group.

[Author reply] Thanks for the comment. We changed the sentence to “The low cost, persistence and high toxicity towards insects make OCPs the ideal candidates in treating soil and plants against various insects. However, OCPs could cause serious health problems for humans.”

Line 197: Ref. 97 describes increases of concentrations in people losing weight (which can be expected for compounds stored in fat), so the ref. does not deal with low-dose chronic exposure being more serious than high-dose short-term exposure. It looks like there is a misunderstanding on the kinetic behavior of slowly eliminating POPs. Because body burden is much more than a daily dose in compounds with a long half-life (e.g. with TCDD body burden at constant intake is about 5000 times the daily dose), it means that a short-term exposure to doses exceeding safety margins set by authorities, do not necessarily increase the body burden much. This was clearly seen in Belgian PCB/dioxin incident as correctly referred to. A correct formulation could be something like: Because slow elimination leads to slow accumulation, the body burden depends more on long-term exposure than occasional exceedances of tolerable daily intake limits. A possible recent reference is Tuomisto et al, A pharmacokinetic analysis … in Toxicology Letters 2016:261:41-48.

[Author reply] Thanks for the suggestion. We added the correct formulation to the manuscript. “Because slow elimination leads to slow accumulation, the body burden depends more on long-term exposure than occasional exceedances of tolerable daily intake limits. [104]”

Line 225: Again, the big threats are more historical than a contemporary problem, although some risks remain and the safety margins are not great.

[Author reply] Thanks for the suggestion. We added the following sentence to the manuscript. “Although the levels of POPs are decreasing, and POPs may not be a big threat to human health now compared to decades ago, the risk remains, and the safety margins still need to be improved. It is worth studying the possible health effects to better protect human health. “

Line 246: The authors probably mean aryl hydrocarbon receptor (AHR). The wording is a bit unclear, do the authors mean that dioxins bind to hormone receptors which is probably not true, but the receptors interact.

[Author reply] Thanks for the comment. The sentence was changed to “Dioxin, furan and dioxin-like PCBs have been reported to influence the activation of transcription factors through binding with the aryl hydrocarbon receptor and interacting with hormone receptors, therefore, affecting the normal hormone function [2, 3]”

Line 279-280: There is a caveat in hot spot studies, because as stated earlier in this MS, dioxin intake is mainly from food, and very little food comes today from local sources, especially very little of animal source food and fish. So it would be very hard to explain why a local hazardous waste site could be crucial, at least there should be thorough analytical evidence.

[Author reply] Thanks for the comment. We changed the sentence to “The study conducted in United States found that exposed to POPs may increase the risk of hypertension. [150].”

Reviewer 3 Report

Guo et al., reviewed the potential contamination sources, detection methods and plausible health outcomes of persistent organic pollutants (POPs). Although POPs are legacy chemicals, still they are of serious concern due to their long-term persistence and long-range transportation across the globe. Overall, the review structure and literature survey are exceptional. This review clearly reflects the overall concepts and updates on POPs in foodstuffs. However, the inclusion of some of the quantitative values such as allowable limits (residual levels and allowable dietary intakes), current detected limits (max), tolerance/threshold limits and risks assessment of POPs chemicals would certainly add more significances.   

Major comments:

Most of these POPs chemicals are halogenated compounds (Br, Cl or F) which make them persistent. So, please include such information in the introduction and any relevant references. Stockholm convention mainly establishes and started banning the “dirty dozen” chemicals. However, the authors did not specify such information anywhere in the text. I strongly believe there are some established “maximum residual levels” values available for these chemicals. Please include such values. Similarly, please include any established NOAEL or reference doses or any toxic endpoints values for at least some of the widely studied chemicals. There are several recent studies reported on the prenatal exposure to POPs and associated risks in children. Please include a couple of relevant references and discussion. \ Except for PFASs, most of the POPs chemicals were measured using GC-MS due to their low or moderate polar nature. Whereas, PFASs related chemicals are always measured with LC-MS/MS methods. So, need such clarity in the section of detection methods. The current version has a very limited discussion on PFASs. Authors should include some of the recent studies on PFASs contamination in foods. Moreover, several studies documented the detection of PFASs in food packing or wrapping materials. Such information needs to be included. PCBs and PBDEs have several congeners. Please, include discussion on the most abundant or detected congeners in foodstuffs. Lack of recent literature: Fishes are one of the major sources for exposure to POPs. So, please include some of the recent studies such as Bedi et al., 2017 (https://doi.org/10.1080/10807039.2017.1421453), Fair et al., 2018 (https://doi.org/10.1016/j.envres.2018.08.001), Abalos et al., 2019 (https://doi.org/10.1016/j.scitotenv.2018.07.371)

Minor comments:

Keywords: Please include keywords “food safety” and “environmental contaminants” Page 6 line 67: Please add abbreviations “EFSA” and “WHO” in respective places. Page 7 lines 105-107: Please add respective abbreviations in parenthesis for each extraction method. Page 15 line 410: “Does not contain clean extract”: It is not a definition for blank samples. Please refer to literature and correct it. Page 15 line 439: It’s “American countries” not America countries.

Author Response

We appreciate the reviewer for the constructive comments. We carefully read the comments and revised our manuscript accordingly. The reviewer’s comments dramatically improved our manuscript. Below we give the point-by-point responses to the reviewer’s comments. We hope our responses satisfy the reviewer.

Most of these POPs chemicals are halogenated compounds (Br, Cl or F) which make them persistent. So, please include such information in the introduction and any relevant references.

[Author reply] Thanks for the comment. We added the following sentence to the manuscript. “Most POPs are halogenated chemicals and the strong bond between carbon and chlorine/bromine/fluorine makes POPs resistant to the environmental degradation including chemical, biological, and photolytic reactions. For those non-halogenated POPs, their stable chemical structures also make them persistent in nature.”

Stockholm convention mainly establishes and started banning the “dirty dozen” chemicals. However, the authors did not specify such information anywhere in the text.

[Author reply] Thanks for the comment. We added the following sentences to the manuscript. “Stockholm Convention requires its parties to take actions to decrease the production, use and releases of POPs on its list. The initial list was established in 2001 including 12 POPs, which are known as “dirty dozen”. By 2019, additional 17 POPs have been added to the list.”

I strongly believe there are some established “maximum residual levels” values available for these chemicals. Please include such values. For PCDD/F and dioxin-like PCBs, For OCP residue, FAO/WHO. Similarly, please include any established NOAEL or reference doses or any toxic endpoints values for at least some of the widely studied chemicals.

[Author reply] Thanks for the suggestion. We added the following sentence to the manuscript. “For PCDD/F and dioxin-like PCBs, the tolerable daily intake was set up by WHO in 2000 to be 1-4 pg TEQ kg-1 body weight [214]; the tolerable weekly intake was recommended by the European Food Safety Authority(EFSA) in 2018 to be 2 pg WHO TEQ kg-1 body weight[215]; and the provisional tolerable monthly intake was set up by Joint Expert Committee on Food Additives (JECFA) of Food Agriculture Organization (FAO) and WHO in 2001 to be 70 pg WHO-TEQ kg-1 body weight[216]. European Union has established the maximum levels for PCDD/Fs and the sum of PCDD/Fs and dioxin-like PCBs in various foodstuffs in 2011.[208, 217] For example, for the sum of PCDD/Fs and dioxin-like PCBs, EU maximum level is 4 pg TEQ/g fat for beef, 5.5 pg TEQ/g fat for milk, 5.0 pg TEQ/g fat for egg, 6.5 pg TEQ/g wet weight for fish except eel and 1.5 pg TEQ/g for compound feed. [208] For DDTs and HCHs, FAO and WHO set up acceptable daily intakes to be less than 10,000 ng/kg body weight/day and 5000 ng/kg body weight/day, respectively. [218] For PAHs, regulatory maximum levels on traded food is 2.0 ug/kg for BaP and 12.0 ug/kg for PAH4 in smoked meat and smoked meat products. [219]”

There are several recent studies reported on the prenatal exposure to POPs and associated risks in children. Please include a couple of relevant references and discussion.

[Author reply] Thanks for the suggestion. We added the following sentence to the manuscript. “Prenatal exposure to POPs not only poses adverse effects on mothers’ health but also on newborns. Recent studies showed that prenatal exposure POPs may be associated with a decrease in birth weight [121, 122], child obesity, increased blood pressure [123] and endocrine disruptive effect [124, 125].”

Except for PFASs, most of the POPs chemicals were measured using GC-MS due to their low or moderate polar nature. Whereas, PFASs related chemicals are always measured with LC-MS/MS methods. So, need such clarity in the section of detection methods.

[Author reply] Thanks for the suggestion. We added the clarification to the manuscript. “Most POPs are semi-volatile, and their polarities are between moderate to non-polar. These physicochemical properties make most POPs well suited to be measured using GC-MS, except for PFASs related chemicals, which are always measured using LC-MS/MS method.”

The current version has a very limited discussion on PFASs. Authors should include some of the recent studies on PFASs contamination in foods. Moreover, several studies documented the detection of PFASs in food packing or wrapping materials. Such information needs to be included.

[Author reply] Thanks for the suggestion. We added the recent studies in the manuscript. “Per- and polyfluoroalkyl substances (PFASs) and their derivatives are a group of chemicals composed of a fluorinated alkyl chain (generally C4-C18) and a hydrophilic functional group.[110] Since these chemicals are waterproof, oil-proof and heat resistant, they are used in a wide range of products and applications such as food packaging, non-stick cookware, cleaning agents, carpet, furniture, coating materials and etc.[111-113]The most investigated PFAS including perfluorooctanesulfonate(PFOS) and perfluorooctanoic acid (PFOA). Perfluorooctanesulfonate (PFOS) is used in paper coating for food contact and perfluorooctanoic acid (PFOA) is a processing aid used in nonstick cookware [114]. PFOS has been detected in many food sources [115-117]. The main sources for PFOAs and PFOSs are fish, seafood, meat, eggs, dairy product and drinking water. [111, 113, 118] Recent studies on fluorinated chemicals in food packaging shows their potentially significant contribution to dietary exposure to PFAS. [118-120]” 

PCBs and PBDEs have several congeners. Please, include discussion on the most abundant or detected congeners in foodstuffs.

[Author reply] Thanks for the suggestion. We added the following sentences in the manuscript. “For PCBs, there are 209 possible congeners and analysis is usually focused on the toxic PCBs including dioxin-like PCBs (PCB 77, 81, 105, 114, 118, 123, 126, 156, 157, 167, 169, 189) and indicator PCBs (PCB-28, 52, 101, 138, 153 and 180).[20, 45, 87-89] For PBDEs, the most detected congeners in foodstuffs include 16 PBDEs (BDE-28, 47, 49, 53, 66, 85, 99, 100, 153, 154, 183, 196, 206, 207, 208 and 209).[45, 90, 91]”

 Lack of recent literature: Fishes are one of the major sources for exposure to POPs. So, please include some of the recent studies such as Bedi et al., 2017 (https://doi.org/10.1080/10807039.2017.1421453), Fair et al., 2018 (https://doi.org/10.1016/j.envres.2018.08.001), Abalos et al., 2019 (https://doi.org/10.1016/j.scitotenv.2018.07.371)

[Author reply] Thanks for the suggestion. We added the recent studies in the manuscript. “Fishes are one of the major sources for exposure to POPs. [43-45]”

Minor comments:

Keywords: Please include keywords “food safety” and “environmental contaminants”

[Author reply] Thanks for the comment. We added “food safety” and “environmental contaminants” in the keywords.

Page 6 line 67: Please add abbreviations “EFSA” and “WHO” in respective places.

[Author reply] Thanks for the comment.  We added abbreviations “EFSA” and “WHO” in the manuscript. 

Page 7 lines 105-107: Please add respective abbreviations in parenthesis for each extraction method.

[Author reply] Thanks for the suggestion. We added the respective abbreviations in the manuscript.

Page 15 line 410: “Does not contain clean extract”: It is not a definition for blank samples. Please refer to literature and correct it.

[Author reply] Thanks for the suggestion. We changed the sentence to the following “A blank sample is an analyte free matrix. It is treated identically to target samples extracted in the same batch, including exposure to all glassware, equipment, solvents, reagents, and internal standards. [193] Blank is used to demonstrate that the system (laboratory, reagents, glassware, etc.) is free of contaminants and ensure the extraction process is under control. For food sample analysis, corn oil is often used as blank samples. [66]”

Page 15 line 439: It’s “American countries” not America countries. 

[Author reply] Thanks for catching this typo. It has been corrected in the revision.

Round 2

Reviewer 1 Report

No further comments.

Reviewer 3 Report

The authors have revised the manuscript carefully by considering all reviewer's queries. The responses and appropriate updates in the manuscript are excellent. Overall, I'm pleased with the present form of the manuscript and recommend it for publication. 

This manuscript is a resubmission of an earlier submission. The following is a list of the peer review reports and author responses from that submission.

Round 1

Reviewer 1 Report

This review may be useful for somebody who is searching for literature on POPs, but it is too mixed bag and should be drastically rewritten before publication.

The review covers a large number of substances with very different properties, and that is its obvious weakness. It is not possible to describe the risks of e.g. dioxins (varying in potency tenthousandfold even among the group) with other POPs very variable in their toxicity and toxicokinetics. One should be much more specific and appreciate that PAHs are not dioxins, PCBs are not pesticides, and PBDEs are not PFOAs. There is no such thing as a common toxicity of POPs.

The second weakness is that effective doses are almost completely ignored. Dose is a central principle in toxicology, because all chemicals, both synthetic and natural, are toxic at some dose level. If PCBs and accompanying PCDFs in Yusho and Yucheng catastrophes caused a number of developmental effects and cancer, it does not imply that thousands of times lower doses would do the same. Much of the evidence of harm is from accidents or occupational toxicity studies, and tells very little on consumer risks.

The third weakness is that the favourable trends in POP intake during the last 50 years are not given fair treatise. POP routes in the environment and in mammals have been thoroughly studied. Breast milk analyses under the auspices of the World Health Organization have shown about tenfold decrease in dioxin and PCB levels (e.g. van den Berg et al, Arch Toxicol. 2017;91:83), and similar or even more drastic decreases have been seen in emissions (e.g. U.S.EPA, 2006,  An inventory of sources and environmental releases of dioxin-like compounds in the U.S. for the years 1987, 1995, and 2000;  Dopico and Gomez, J  Air & Waste Manag  Ass. 2015;65:1033), and similarly the most harmful pesticides have been abandoned many years ago, and human levels have decreased. So this MS is woefully outdated in the present form. It is possible that in developing countries a lot remains to be done especially in educating proper pesticide use, but the favourable development in many countries should be appreciated and the remaining contemporary problems expressed specifically.

Many of the population studies are ecological observational studies which are notoriously poor in showing causal relation. Diabetes is one example where a wide variety of completely different chemicals has been implicated, but the obvious confounding factors of obesity and animal source food as a common determinant for both long-acting chemicals and diabetes have been poorly controlled.

It is difficult to know, if anybody is able to write a balanced review on such a wide scope without being too general, but the authors are encouraged to be much more specific, and if necessary, to divide the review in manageable pieces without having to fall into the trap of too wide generalizations that simply cannot be true and plausible for all POPs or even most of them.

A few examples of points to consider (not exhaustive, and the analytical methods, which are a collection of everything, have not been touched):

Lines 14-17: List of health problems is a swiping generalisation that is mostly not true, even if it can be true in some specific cases, mostly at high exposures and not ”at trace level”.

Line 22: increasing amounts of residues are not true, in fact POPs are decreasing in most countries, the peak was 1970s in most cases. PBDEs and PFOS were introduced later, but be specific

Line 27: Concerns are not recent, they started after Rachel Carson’s Silent spring in 1962, and serious scientific focus has been given to the issue at least since 1970s, and even the Stockholm convention is from 2001. The text gives a wrong impression that nothing has happened.

Line 37 and Table 1: PAHs are not persistent organic pollutants

Line 43: POPs in general are poorly water soluble, but many of them are not highly fat soluble either (e.g. dioxins), although lipid/water coefficient is high

Line 45: Vapor pressure of some POPs is very low and they are not transported by evaporation, so e.g. PCBs may evaporate and are transported to arctic regions, but dioxins are not; be careful with the differences

Line 48-50: what is a trace level? Some dioxins are certainly potent and toxic levels are in picogram range per gram, but many PCBs are quite inert; toxaphene is acutely toxic, but DDT in mammals is not. Be specific, many of the health problems listed are based on poorly controlled studies and are not generally true for many POPs.

Lines 54-55: The released POPs are able to do many things, but do they? The levels are decreasing, and in most countries they are well below safety limits (usually set with ample safety margins) and often even below detection limits, even if the chemists do their best to develop ever more sensitive analytical methods. This also applies to pesticides (line 59).

Lines 68-71: Risk assessment cannot be based on analytical capabilities, because there are two essential parts, dose (exposure) and effect, especially dose-effect relationship. At present the analytical capabilities are quite adequate to assess the amounts of POPs in food or feed, and the greatest difficulty is in assessing what is the dose level that causes a risk. By and large only dioxins require pg/g capabilities.

Line 94: number of POPs in a sample is totally uninformative, and if the chemist is good enough, trace amounts of tens or hundreds of POPs will be found in a food sample.

Table 2: surprisingly the most important source of dioxins is not listed: fish. Fish is much more important than vegetable oils.  This illustrates haphazard selection of literature.

Lines 156-163: It is not rational to bundle up Yusho and Yucheng with Belgian or Irish contamination incidences. Yusho and Yucheng caused extremely serious poisonings, but also concentrations were extremely high (intake 100,000 times higher than the present background intake, Masuda, Chemosphere. 1996;32:583). In the Belgian incident the concentrations in the population did not increase, and no health consequences have been noted (Debacker et al., Chemosphere. 2007;67:S217). The same applies for the Irish incident (Pratt et al, ref. 70).

Lines 182-: Concentrations of many POPs were tenfold in 1970s, so how could they be a big threat to human health now? Some health effects are possible although controversial, and many of the health consequences in table 3 have been rather poorly documented. So far the best evidence is from animal experiments. Developmental effects probably have the lowest safety margins at least with dioxins. Pesticides are one of the most unlikely causes of any toxic effects or illnesses in well-organized societies, although e.g. in Africa they cause a number of poisonings.

Line 189: Endocrine disruption is one of the most problematic effects to show unambiguously. There are more claims than proven facts. Although some POPs clearly affect endocrines, a claim that POPs in general are endocrine disruptors is probably not true. Even if true there is a big discussion on the dose at which the effects are plausible.

Line 224: ecological studies around industrial or waste sites are a prime example of weak evidence, especially if based on interviews. In a review much of the effort should be given to assessing the credibility of the methods and the likelihood of confounding.

Lines 231-238: the difficulty with diabetes studies is that a number of compounds with very different mechanisms of action have been claimed to have similar effects. It seems that the only common denominator is a long half-life, and this again suggests that diet has a confounding role in increasing simultaneously both POP intake and calories. Obesity is the best characterized risk factor of type 2 diabetes.

Line 392: Yusho and Yucheng were serious accidents, and they do not illustrate very well the need of legislation, and e.g. the Belgian incident was a clear criminal act. Of course there is a need to prevent accidental or deliberate contamination, but it is more important for population health to assure that a continuous release (e.g. from waste incineration) to the environment of toxic compounds is minimized and pesticide use is rational in the same manner as the use of medicines is required to be rational.

Reviewer 2 Report

General

The paper is an incomplete and fragmentary review on the important class of Persistent Organic Pollutants (POPs) in food. The authors have only a poor knowledge on the various international and national regulatory frameworks. In particular, the UNEP Stockholm Convention on POPs is not recognized in its effect on reducing the environmental contamination with POPs. A comprehensive overview on the various national and international regulatory frameworks was published by  A. D. K. Abelkop, J. D. Graham and T. V. Royer (2015): Persistent, Bioaccumulative, and Toxic (PBT) Chemicals: Technical Aspects, Policies, and Practices, CRC Press, Boca Raton, FL. The origin and evolution of PBT and POP criteria were reviewed by M. Matthies, K. Solomon, M. Vighi,  A. Gilman and J. V. Tarazona (2016) Environ Sci: Processes & Impacts 18, 1114–1128.

The major source for food and feed contamination with POPs is their slowly declining global environmental reservoir due to the large amounts emitted in the past, which is completely neglected. POPs are semivolatile compounds, i.e. they are transferred between air on the one hand and water, soil and plants on the other hand. Uptake of POPs from air into plant foliage is very effective and the major source of feed contamination and subsequent meat and milk contamination.

Improvement of POPs regulations are proposed but not specified.

Specific

Line 29  Add Stockholm Convention (SC) to the references.

Line 30  ‘...may contain chlorine, bromine, or fluorine in their structure.’ This is not a criterion for POPs. PAHs are a counter example. According to the UNEP Stockholm Convention, POPs are defined by their persistence, long-range transport potential, bioaccumulation and toxicity, not by a particular structure.

Line 37ff Tab. 1 displays only the 11 initial POPs but not all 16 new POPs.

Line 39  see my comment on l. 30

Line 45  Long-range transportability (LRT) is a specific property of POPs. Other chemicals with persistent, bioaccumulative and toxic properties, but no LRT are classified as PBTs and regulated under various national legislations. This distinction should be mentioned in the paper.

Line 51  ‘more than 90% .... of contaminated food’. Provide reference.

Line 65  SC is the most prominent, legal binding international framework, not an example. However, maximum permissible values in food are issued by national legislations.

Line 77  pH, not PH.

Line 91ff              This chapter completely neglects the large environmental reservoir from past releases as the main source of POPs contamination of food.

Line 120ff            Production and use of DDT is accepted for disease vector control according to SC, Annex B.

Line 163               After ‘primary sources of’ insert ‘single, extremely high’.

Line 186, Tab. 3   PAHs and many new POPs are missing.

Line 369               Delete first sentence.

Line 370               Replace ‘are required’ by ‘ have been issued’.

Line 375               Replace ‘Some’ by ‘All’.

Line 376f             See general comments. Moreover, maximum permissible values are the instrument for controlling hazardous chemicals (not only POPs) in food, but are not sufficiently discussed.

Line 380               See comment on line 120ff.

Line 390               Provide reference for EC regulation 159/2011.

Line 392               Provide references for the two accidents.

Line 418               Delete the whole sentence ‘Rapidly growing...into the environment’.

Line 419f             ‘In addition, .... Reference is needed for this statement.

Line 420f             ‘Therefore, the strong regulation at global level is needed to reduce the use of POPs and ultimately eliminate their release.’ Should the SC replaced by a new Convention? Or should it improved? And how?

Line 422f             ‘The United Nations Environment Programme defines criteria for new POPs candidates in terms of their persistence, long-range transport, bioaccumulation and toxicity.’ Yes, indeed in Annex D of the Stockholm Convention. Altogether, 16 new POPs have been classified by applying these criteria since 2001 and other chemical have been put on a list of POP candidates. Should these criteria be changed?

Line 435               ‘...to improve the POPs regulations’. Which regulations? What improvements?